# Effect of Insurance-Related Factors on the Association between Flooding and Mental Health Outcomes

**DOI:** 10.3390/ijerph16071174

**Published:** 2019-04-02

**Authors:** Ranya Mulchandani, Melissa Smith, Ben Armstrong, Charles R Beck, Isabel Oliver

**Affiliations:** 1Field Epidemiology, Field Service, National Infection Service, Public Health England, Bristol BS1 6EH, UK; ms8666@bristol.ac.uk (M.S.); Charles.Beck@phe.gov.uk (C.R.B.); Isabel.Oliver@phe.gov.uk (I.O.); 2NIHR Health Protection Research Unit in Evaluation of Interventions, University of Bristol, Bristol BS8 2BN, UK; 3Bristol Veterinary School, University of Bristol, Bristol BS40 5DU, UK; 4NIHR Health Protection Research Unit in Environmental Change and Health, London School of Hygiene and Tropical Medicine, London WC1H 9SH, UK; Ben.Armstrong@lshtm.ac.uk; 5Public Health England, London SE1 8UG, UK; Thomas.Waite@phe.gov.uk; 6Population Health Sciences, Bristol Medical School, University of Bristol, Bristol BS8 2PS, UK

**Keywords:** mental health, flooding, natural disasters, insurance

## Abstract

Floods are a significant public health problem linked with increased psychological morbidity. We aimed to investigate the effect of insurance-related factors on the association between flooding and probable mental health outcomes. We performed a secondary analysis of cross-sectional survey data from the English National Study of Flooding and Health (NSFH) collected two years after an initial flooding event in 2013-14. Our analysis focused on 851 respondents who experienced flooding or disruption. Multivariable logistic regression models were run for each exposure group. Among those whose homes had been flooded, not having household insurance was associated with increased odds of all outcomes compared to those with household insurance, significantly so for post-traumatic stress disorder (PTSD) (aOR 4.31, 95% CI 1.31–14.20). Those who reported severe stress due to insurance issues had increased odds of probable depression (aOR 11.08, 95% CI 1.11–110.30), anxiety (aOR 4.48, 95% CI 1.02–19.70) and PTSD (aOR 7.95, 95% CI 2.10–30.1) compared to those reporting no/mild stress. The study suggests there is increased psychological morbidity amongst the uninsured and those who report feeling severe stress as a result of insurance issues associated with flooding. Services should be prepared to support communities through insurance processes, to reduce probable mental health morbidity following a flood event.

## 1. Introduction

Flooding is the most common global natural hazard, with its frequency and intensity anticipated to increase as a result of climate change [1]. Long-term health problems, in particular adverse mental health outcomes, have been linked to the experience of flooding, including elevated levels of depression, anxiety and post-traumatic stress disorder (PTSD) in affected populations [2,3,4]. 

In the winter of 2013–2014, widespread flooding occurred in many parts of the UK, with total economic damages estimated at £1.3 billion [5]. The government responded with an innovative scheme called “Repair and Renew”, accessed through local authorities. The scheme gave affected homeowners the opportunity to apply for a £5000 grant to carry out home improvements that would increase flood resilience, in addition to any pre-existing flood and home insurance cover [6]. 

Public Health England’s (PHE) response to the floods included the establishment of the National Study of Flooding and Health (NSFH) working with academic partners to investigate the long-term health impacts of flooding and related ongoing disruptions on mental health. The NSFH aims to support preparedness and guide appropriate public health responses in the event of future flooding events. 

The NSFH has identified an increased burden of probable psychological morbidity in people whose homes were flooded or whose lives were affected by flooding compared to unaffected persons [7,8]. The NSFH has also found a higher prevalence of adverse mental health outcomes in those who were displaced from their homes as a result of flooding, in particular those with no prior warning of evacuation [9]. 

Secondary stressors such as dealing with insurance issues after flooding have been associated with psychological morbidity [10,11]. Qualitative studies have shown that flooded respondents felt that disputes with insurance companies had created and worsened psychological health issues [12], and many reported that problems with flood insurance cover was a key reason for their stress [13]. However, these factors have not previously been quantified [14] and it is not known to what extent insurance issues are a driver for the association between flooding and psychological morbidity.

This study quantified the association between insurance-related factors (including status of insurance during the flooding event, experience of insurance-related issues, reports of stress levels due to insurance-related issues, claims submitted and level of repayment) and psychological outcomes following a flood event. 

## 2. Materials and Methods 

We conducted a secondary analysis of cross-sectional survey data from the NSFH collected two years following floods (December 2013–March 2014). The study design, sampling and recruitment are described elsewhere [8]. Eligible participants (individuals aged 18 years and over) were recruited using a bespoke questionnaire, including questions on probable mental health outcomes, insurance status at the time of flooding, any subsequent claims for insurance or local/national government grants and level of insurance claim repayment. Probable mental health outcomes were measured at year two using validated tools and were defined as the presence or absence of probable depression, anxiety and PTSD measured by PHQ-2 (threshold ≥ 3) [15], GAD-2 (threshold ≥ 3) [16] and PCL-6 (threshold ≥ 14) [17] scales respectively. 

Participants were split into three categories at year one, based on their experience of flooding: flooded (entry of water into any liveable room of the home); disrupted (“life disrupted” by flooding, including reported interruption and/or loss of access to household and community services, but no entry of water into a liveable room of the home); and unaffected by flooding. Our analysis was restricted to respondents who consented to follow up after year 1, experienced flooding or disruption (i.e., excluding the unaffected group) and returned a questionnaire at year two with at least one question answered. Any respondents with missing values for exposure or who reported new episodes of flooding since 2013/2014 were excluded.

We performed descriptive statistical analyses of respondents’ demographics, exposure to flooding, insurance status at time of flooding, insurance claim repayment status and insurance-related issues. Persons with insurance included those with buildings and/or contents insurance. We described reports of dealing with “insurance-related issues” due to flooding, reported stress levels and probable mental health outcomes of depression, anxiety and PTSD due to these issues. 

Logistic regression models were run for each of the disrupted and flooded exposure groups to clarify uncertainty in the association between these insurance-related variables (insurance claim status, “Repair and Renew” grant claim status, reporting of “insurance-related issues”, reporting insurance-related stress, insurance repayment and grant repayment statuses) and probable mental health outcomes. Multivariable logistic regression models were also performed for each exposure group to adjust for a priori potential confounders: age group, sex, ethnic group, pre-existing illness, deprivation score (index of multiple deprivation (IMD)), marital status, education and local authority. 

Data were managed and cleaned using Stata version 13.1 (StataCorp LLC, College Station, TX, USA) and statistical analyses were conducted in R version 3.4.1 (R Foundation for Statistical Computing, Vienna, Austria). 

## 3. Results

Questionnaires were returned from 1064 households, with an overall response rate of 76.0% out of 1408 people who had consented to follow up [7]. Following exclusions due to being duplicates (*n* = 20), having been affected by new episodes of flooding (*n* = 18), not providing sufficient information to be assigned an exposure category (*n* = 38) or being classified as “unaffected” (*n* = 137), 851 participants remained in this analysis (512 disrupted and 339 flooded). In the disrupted group, 4.6% were classified as having probable depression, 7.2% of participants as having probable anxiety and 9.9% as having probable PTSD. In the flooded group, 11.9%, 15.2% and 27.0% of participants were classified as having probable depression, anxiety and PTSD respectively. The prevalence of probable mental health outcomes, stratified by insurance status can be found in Table 1. 

The majority of participants (425 (83.0%) in the disrupted group and 320 (94.4%) in the flooded group) reported having household insurance prior to the flooding episode. Proportions of uninsured differed significantly for those reporting pre-existing illness and education level, but not by sex, age, marital status or ethnic group. For those reporting pre-existing illness 18.0% (39/216) were uninsured compared to 10.8% (82/674) of those who reported no pre-existing illness (*p* = 0.03). Of those with no formal education (no school-level qualifications), 21.9% (23/105) were uninsured, compared to 9.9% (36/363) of those with below degree level qualification and 9.4% (36/381) with a degree or above (*p* < 0.001). 

Forty-seven participants (9.8%) in the disrupted group and 280 participants (87.5%) in the flooded group had submitted an insurance claim following this flooding event while 23 (5.2%) participants from the disrupted group and 198 participants (77.0%) from the flooded group had applied for a grant through the “Repair and Renew” scheme. Fifty-six percent of participants in the flooded group reported “dealing with issues” as a result of their insurance claim, compared to only 12.8% from the disrupted group. Of all those who reported dealing with insurance-related issues, 22.1% reported no/mild stress, 52.3% reported moderate stress and 25.6% reported severe stress in the flooded group; 23.7%, 55.9% and 20.3% reported the respective stress levels in the disrupted group. 

A comparison of prevalence and odds ratios (ORs) indicated that not having insurance at the time of flooding was associated with increased odds of all adverse outcomes, significantly for depression (OR 3.45, 95% CI 1.02–10.13) and PTSD (OR 3.26, 95% CI 1.19–9.13) in the flooded group, compared to those with insurance at the time of flooding. Adjustment for confounders did not change our OR estimations largely, but did increase their uncertainty. Not having insurance at the time of flooding was however still significantly associated with increased odds of PTSD (aOR 4.31, 95% CI 1.31–14.20) among the flooded respondents (Table 2). 

The prevalence for the majority of adverse mental health outcomes was higher in those who submitted an insurance claim, reported dealing with “insurance-related issues” and reported stress due to insurance issues for disrupted or flooded groups (Table 3 and Table 4). For the flooded group (Table 3), those who reported experiencing “insurance-related issues” due to flooding had 2.5 times the odds of PTSD (aOR 2.54, 95% CI 1.10–5.85) compared with those who did not report experiencing any issues. The prevalence of anxiety was also raised in this group, but not significantly. Reporting of severe stress as a result of insurance issues due to flooding had significantly increased odds for all three mental health outcomes—depression (aOR 11.08, 95% CI 1.11–110.30), anxiety (aOR 4.48, 95% CI 1.02–19.70) and PTSD (aOR 7.95, 95% CI 2.10–30.10) compared to reporting no/mild stress (Table 3). There were broadly similar patterns in the disrupted respondents (Table 4), but the lower prevalence in general in this group led to greater uncertainty such that no association was significant. 

## 4. Discussion

This study found that those who were flooded and did not have household insurance at the time of flooding were more likely to experience probable depression, anxiety, and PTSD after two years following the original flooding event than those who did have insurance, significantly so for PTSD. Our study observed higher proportions of uninsured individuals in those reporting pre-existing illness and lower education levels. Those who were flooded and reported feeling stressed due to “insurance related issues” were at significantly higher odds of developing all three probable mental health outcomes examined (depression, anxiety and PTSD) suggesting that insurance-related factors contribute to the psychological morbidity associated with flooding. 

Our study fits within the existing literature on the association between flooding and mental health outcomes [2,3,18,19,20,21]. Two studies following floods, have identified reports of “difficulties with compensation” including applications for insurance pay outs or state grants delayed or denied, a lack of understanding with respect to application processes, and often insufficient help or advice available in making claims [13,22]; however, no quantitative studies were identified investigating the association between such insurance issues and psychological morbidity following flooding. 

To our knowledge, this is the first study that describes the role of insurance status and related-issues on psychological morbidity in the context of flooding using a quantitative approach. The study also investigated reported stress relating to insurance-related factors and its associations with the odds of probable mental health outcomes after flooding. 

The strengths and limitations of the study design, the outcome measurement tools used and the general nature of the NSFH have been discussed elsewhere [7,8]. Specific to this study, the low response to questions on amount paid out for both insurance and “Repair & Renew” claims limited our ability to analyse the association between levels of payment (i.e., whether the claim had been unsuccessful or not) and odds of probable mental health outcomes, and does not imply a true lack of effect.

There were no significant associations seen for any insurance-related variable with any mental health outcome for those classified as disrupted; however, this could be due to our lack of information on the size of claims by this group. For example, disrupted respondents may have submitted smaller claims, and therefore have been less dependent on the outcome of their claim and consequently less impacted by the anxieties surrounding the overall claim process. Our measurement of stress may have introduced an element of recall bias, however self-reported stress is a valid measure and important for our understanding on respondents’ experiences. Our questionnaire did not explicitly characterise the “insurance-related issues” and this was therefore open to interpretation by the respondent. It would be helpful to understand this issue better to help guide future support mechanisms. 

We acknowledge the possible role of other possible confounders related to insurance policies (e.g., whether people carried sufficient insurance) and other secondary stressors, which were not measured within our study, such as the availability of social support. As individual socio-economic status or income information was not available, IMD scores were used as a proxy. In addition, there might have been other factors which could increase the probability of both not having insurance, and of developing mental issues (e.g., financial or prior housing-related issues including past flooding experience). In future studies it would be worth investigating these further. Finally, further research is needed to further understand the associations between individual economic status, level of damage, insurance claims and mental health.

## 5. Conclusions

The study emphasises the importance of having home insurance on mitigating the psychological impacts of flooding. In particular, it would be important to identify those most likely to be uninsured (such as those with pre-existing illness or those with no formal education as observed in our study) to better understand and address the underlying reasons for their lack of insurance. 

The study also suggests that there is increased psychological morbidity amongst those who deal with insurance-related issues post-flooding, which is important knowledge for support services such as those provided by local authorities, as well as voluntary and community organisations in areas affected by flooding. Our study suggests that further support is needed to help people navigate applications and that insurance companies should provide effective, straightforward claims processes to reduce probable mental health morbidity associated with insurance-related issues post flooding. 

## Figures and Tables

**Table 1 ijerph-16-01174-t001:** Crude proportions of mental health outcomes by those who were insured and uninsured ^±^.

Outcome	Affected Cohort *	Disrupted Insured	Uninsured	Flooded Insured	Uninsured
Probable depression	57/762 (7.4%)	16/386 (4.1%)	5/74 (6.8%)	31/285 (10.8%)	5/17 (29.4%)
Probable anxiety	79/759 (10.4%)	27/384 (7.0%)	6/73 (8.2%)	41/285 (14.4%)	5/17 (29.4%)
Probable PTSD	129/774 (16.7%)	35/392 (8.9%)	11/75 (14.7%)	74/290 (25.5%)	9/17 (52.9%)

^±^ Footnote: The crude prevalence of mental health outcomes presented here is not exactly comparable to those in the previously published in Table 2 of the paper using the same data by Jermacane et al., BMC Public Health (2018) for flooded and disrupted overall. The Jermacane paper included in denominators individuals who had responded to some but not all mental health questions, but in the current paper those subjects were omitted, in line with the approach of Waite et al. and BMC Public Health (2017). * disrupted and flooded participants.

**Table 2 ijerph-16-01174-t002:** Crude and adjusted odds ratios (ORs) or psychological morbidity amongst disrupted and flooded participants, comparing those who did not have insurance at the time of flooding with those who did.

Outcome	Crude OR (95% CI)	aOR * (95% CI)
Probable depression		
Disrupted	1.71 (0.53–4.57)	1.31 (0.35–4.80)
Flooded	3.45 (1.02–10.13)	3.14 (0.84–11.70)
Probable anxiety		
Disrupted	1.21 (0.43–2.87)	1.38 (0.47–4.04)
Flooded	2.51 (0.74–7.26)	2.92 (0.81–10.49)
Probable PTSD		
Disrupted	1.77 (0.81–3.57)	1.22 (0.48–3.08)
Flooded	3.26 (1.19–9.13)	4.31 (1.31–14.2) ^

* adjusted odds ratios are adjusted for age group, sex, ethnic group, pre-existing illness, deprivation score, marital status, education and local authority. ^ *p* < 0.05.

**Table 3 ijerph-16-01174-t003:** Association between probable mental health outcomes and insurance-related factors for flooded respondents.

Explanatory Variable	Outcome								
	Depression			Anxiety			PTSD		
	Prevalence % (*N*)	aOR	(95% CI)	Prevalence % (*N*)	aOR	(95% CI)	Prevalence % (*N*)	aOR	(95% CI)
Insurance claim made									
No	14.2 (35)	ref	ref	13.9 (36)	ref	ref	16.7 (36)	ref	ref
Yes	10.8 (250)	1.07	(0.26–4.42)	14.8 (250)	1.03	(0.27–3.94)	27.0 (255)	1.57	(0.49–5.06)
Repair and Renew grant application made									
No	11.5 (52)	ref	ref	10.9 (55)	ref	ref	18.5 (54)	ref	ref
Yes	15.6 (173)	2.58	(0.75–8.83)	19.2(177)	3.05	(0.95–9.80)	31.3 (176)	1.98	(0.84–4.70)
Experienced dealing with insurance-related issues due to flooding									
No	12.4 (121)	ref	ref	11.5 (122)	ref	ref	20.0 (125)	ref	ref
Yes	11.8 (161)	1.85	(0.98–3.51)	18.1 (160)	1.19	(0.49–2.87)	32.1 (165)	2.54	(1.10–5.85)
Amount of stress due to insurance issues									
No/mild stress	7.3 (41)	ref	ref	14.3 (42)	ref	ref	16.3 (43)	ref	ref
Moderate stress	9.4 (85)	4.39	(0.45–42.52)	14.5 (83)	0.80	(0.19–3.39)	29.8 (84)	1.54	(0.45–5.27)
Severe stress	25.6 (43)	11.08	(1.11–110.3)	32.6 (43)	4.48	(1.02–19.7)	62.8 (43)	7.95	(2.10–30.1)

Reference group is highlighted in the table for each individual factor. Adjusted odds ratios are adjusted for age group, sex, ethnic group, pre-existing illness, deprivation score and education.

**Table 4 ijerph-16-01174-t004:** Association between probable mental health outcomes and insurance-related factors for disrupted respondents.

Explanatory Variable	Outcome								
	Depression			Anxiety			PTSD		
	Prevalence % (*N*)	aOR	(95% CI)	Prevalence % (*N*)	aOR	(95% CI)	Prevalence % (*N*)	aOR	(95% CI)
Insurance claim made									
No	4.1 (394)	ref	ref	9.8 (393)	ref	ref	8.7 (355)	ref	ref
Yes	4.7 (43)	0.73	(0.08–6.80)	6.9 (41)	1.35	(0.36–5.10)	17.1 (47)	2.32	(0.81–6.68)
Repair and Renew grant application made									
No	4.2 (380)	ref	ref	6.9 (379)	ref	ref	8.0 (287)	ref	ref
Yes	0.0 (20)	0	(0–∞)	5.3 (19)	1.05	(0.12–9.33)	19.0 (21)	2.40	(0.58–0.85)
Experienced dealing with insurance-related issues due to flooding									
No	4.3 (375)	ref	ref	7.0 (372)	ref	ref	9.3 (386)	ref	ref
Yes	8.6 (58)	3.92	(0.92–17)	12.0 (58)	1.76	(0.61–5.06)	17.5 (57)	2.23	(0.87–5.76)
Amount of stress due to insurance issues									
No/mild stress	0.0 (11)	ref	ref	0.0 (10)	ref	ref	9.0 (11)	ref	ref
Moderate stress	13.3 (15)	∞	(0–∞)	6.7 (15)	1.46	(0.07–31.86)	14.3 (14)	1.80	(0.09–35.15)
Severe stress	0.0 (7)	1.45	(0–∞)	28.5 (7)	11.37	(0.31–414.2)	28.6 (7)	2.19	(0.07–64.4)

Reference group is highlighted in the table for each individual factor. Adjusted odds ratios are adjusted for age group, sex, ethnic group, pre-existing illness, deprivation score and education.

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
