# Peer review of "Effect of Insurance-Related Factors on the Association between Flooding and Mental Health Outcomes"

_ijerph, 2019, doi:10.3390/ijerph16071174_

Round 1
Reviewer 1 Report
The paper is straight forward in its objective to look at the association between difficulty with insurance after English flood and psychological disorders. The main comment is about what is meant by insurance. There is no mention of home ownership but it would be assumed that people that own their homes would have insurance and others who would not have home insurance but perhaps some sort renter's insurance. It is interesting to note that the uninsured were primarily those with pre existing illness or low education. These variables may be surrogates for income or home ownership which were not reported or measured in the study and could be confounded with insurance issues as associated with mental health issues. Another point that is unclear is whether people who applied for grant through Repair and Renew were one's without insurance. In addition, how much it would cost to fix the homes would be another important variable to include in the analysis, although it does seem be enough to have used insurance related issues as the predictors.
Reviewer 2 Report
Dear authors,
Thank you for writing this paper on flooding, mental health effects and associations with insurance factors.
In general, I find your research question important and the paper add value aspects and knowledge to the field.
Some notes and suggestions to further improve your paper (please ignore any English language concerns, when English is not my first language):
Keywords: Add insurance
Results: I wonder if any of the study participnats were evacuated from their home, for a limited or longer period? That has been found to be associated with mental health problems after other natural events.
Discussion:
Your statement and summery of the findings in the beginning of the discussion is clear; insurance-related factors 173 contribute to the psychological morbidity associated with flooding.
Please, discuss the potential cause for these diagnoses of other events or processes than the flooding. It might also be possible that these (or at least some) had these problems before the flooding.
All the mentioned mental problems are more commonly reported in low income, less educated people. The reasons for not having a housing insurance might also be related to these circumstances, and needs to be considered as a confounder as well.
Also, the factor of social support, that has been found to “buffer” traumatic experiences from people who have been losing their homes in wild fires, is not part of your study, and maybe, should be mentioned as a limitation.
We also know from studies on mental health after traumatic events that many people do recover under the second year after the event. Is it possible to follow up these study participants for another year, this might add value data to the field of psychosocial recovery process.
